# The Acquisition of Colistin Resistance Is Associated to the Amplification of a Large Chromosomal Region in *Klebsiella pneumoniae* kp52145

**DOI:** 10.3390/ijms22020649

**Published:** 2021-01-11

**Authors:** María Blanca Sánchez, Alicia Sánchez-Gorostiaga, Trinidad Cuesta, José Luis Martínez

**Affiliations:** Departamento de Biotecnología Microbiana, Centro Nacional de Biotecnología, CSIC, Darwin 3, Cantoblanco, 28049 Madrid, Spain; blanca.sanchez@imdea.org (M.B.S.); asanchez@cnb.csic.es (A.S.-G.); tcuesta@cnb.csic.es (T.C.)

**Keywords:** *Klebsiella pneumoniae*, colistin resistance, gene amplification

## Abstract

The appearance of carbapenem-resistant *Klebsiella pneumoniae* has increased the use of colistin as a last-resort antibiotic for treating infections by this pathogen. A consequence of its use has been the spread of colistin-resistant strains, in several cases carrying colistin resistance genes. In addition, when susceptible strains are confronted with colistin during treatment, mutation is a major cause of the acquisition of resistance. To analyze the mechanisms of resistance that might be selected during colistin treatment, an experimental evolution assay for 30 days using as a model the clinical *K. pneumoniae* kp52145 isolate in the presence of increasing amounts of colistin was performed. All evolved populations presented a decreased susceptibility to colistin, without showing cross-resistance to antibiotics belonging to other structural families. We did not find any common mutation in the evolved mutants, neither in already known genes, previously known to be associated with the resistance phenotype, nor in new ones. The only common genetic change observed in the strains that evolved in the presence of colistin was the amplification of a 34 Kb sequence, homologous to a prophage (Enterobacteria phage Fels-2). Our data support that gene amplification can be a driving force in the acquisition of colistin resistance by *K. pneumoniae*.

## 1. Introduction

Polymyxins are polycationic peptides that were discovered in 1947 and used until the mid-1990s. However, the fact that they present some degree of nephrotoxicity and neurotoxicity made their use after those dates restricted to just a few types of infections [1,2]. Nevertheless, the increased prevalence of antibiotic-resistant Gram-negative pathogens and the lack of new antibiotics has led to a resurrection of antibiotics such as polymyxins, which are still useful, despite their side effects, when few other therapeutic options are available [3]. In this regard, polymyxins are active against relevant Gram-negative pathogens such as *Pseudomonas aeruginosa*, *Acinetobacter baumannii*, *Klebsiella* spp. and *Escherichia coli*, and other *Enterobacterales* [4].

The primary target of colistin (polymyxin E) is the lipopolysaccharide (LPS) of the outer membrane (OM) of Gram-negative bacteria. Colistin acts as a detergent-like compound, causing the displacement of Ca^+2^ and Mg^+2^ divalent cations from the phosphate groups of LPS that act as membrane stabilizers. This displacement increases membrane permeability and finally renders the disruption of the OM [5].

Bacteria employ several strategies to avoid colistin toxicity. The main resistance mechanisms known so far include modifications of lipid A with 4-amino-4-deoxy-L-arabinose (Ara4N) or phosphoethanolamine (PEtN), loss of LPS, formation of capsules and overexpression of efflux pumps or outer membrane proteins [4,6,7]. Although most of these resistance mechanisms involve the mutation of chromosomal genes, plasmid-encoded colistin resistance genes are particularly relevant. The main mobile colistin resistance gene is *mcr-1*. This gene, which encodes a phosphoethanolamine transferase, was firstly found in *E. coli* and *K. pneumoniae* plasmids in China [8,9] and is now present widespread [10]. More variants of the *mcr* gene have emerged (*mcr-2* to *mcr-9*), but *mcr-1* remains the most prevalent one [11].

*K. pneumoniae* is a Gram-negative bacterial pathogen, which is a common cause of nosocomial infections and is included in the group of ESKAPE pathogens (acronym used to describe the following Gram-positive and Gram negative bacteria: *Enterococcus faecium*, *Staphylococcus aureus*, *Klebsiella pneumoniae*, *Acinetobacter baumannii*, *Pseudomonas aeruginosa*, and *Enterobacter* spp.) [12] due to their resistance to several antibiotics. Carbapenems have been used for the treatment of *K. pneumoniae* infections due to the increased prevalence of extended-spectrum-lactamases (ESBL)-producing isolates [10]. Nevertheless, carbapenem-resistant *K. pneumoniae* (CR-KP) strains are also becoming widespread, making the use of other antibiotics, such as colistin, needed in the treatment of infections by resistant *K. pneumoniae* strains. The use of colistin for treating *K. pneumonia* infections has increased the selection of colistin resistant strains. Besides the aforementioned acquisition through horizontal gene transfer (HGT) of colistin resistance genes, in several cases, together with ESBLs, mutations in different genes have been identified in colistin-resistant *K. pneumoniae* strains. Modification of lipid A has been related with mutations in *phoPQ*, *pmrAB*, *arnBCADTEF* (also known as *pmrHFIJKLM*), *pmrD*, *pmrC*, *mgrB* and *crrAB* [13,14,15,16,17,18], while mutations in *lpxC*, *lpxA* and *lpxD* are associated with the loss of the LPS [19]. Inactivation of *mgrB* has been found to be the most prevalent mutational colistin-resistance mechanism among those so far described [20,21]. Other mechanisms of resistance include changes in the capsule polysaccharide [22] and the overexpression of efflux pumps as AcrAB and KpnEF or the outer membrane protein OprH [7,23,24]. The relevance of the different types of mutations may depend on the bacterial species involved. In this regard, it is worth mentioning that mutations in *lpx* genes are more common in *A. baumannii* while mutations in *phoPQ*, *pmrAB* and others are more common in *Enterobacterales* [11].

Molecular epidemiology of antibiotic resistance is based on the analysis of the already known antibiotic resistance genes/mutations. Experimental evolution approaches have been demonstrated to be valuable tools for analyzing the selection of genetic changes that may contribute to the emergence of antibiotic resistance, even before they are detected in clinical settings [25,26,27,28,29,30]. Among them, gene amplification is a phenomenon that has not been analyzed in detail. In the current work, using experimental evolution approaches, we detected that the amplification of a region of approximately 34 Kb. of the *K. pneumoniae* genome seems to be responsible for the acquisition of reduced-colistin-susceptibility phenotype since no further common mutations in genes previously described to be involved in the acquisition of colistin resistance by bacterial pathogens were found in the evolved populations.

## 2. Results

### 2.1. The Acquisition of Colistin Resistance Does Not Involve Cross-Resistance to Other Antibiotics

The strain *K. pneumoniae* kp52145 [31], also known as CIP52.145 [32], belongs to the *K. pneumoniae* KpI group that includes the majority of strains associated with human infection, including several multidrug-resistant and hypervirulent clones [33]. It is then a good model for searching for novel potential mechanisms of antibiotic resistance. Towards this goal, *K. pneumoniae* kp52145 was submitted to experimental evolution under increasing concentrations of colistin as described in Methods. After 30 days of evolution in the presence (or absence, in control populations) of supra-inhibitory colistin concentrations, samples were subcultured without antibiotic to discard false or transient antibiotic resistance, and colistin Minimal Inhibitory Concentrations (MICs) were determined by double dilution. The MIC of the wild-type strain was 2 µg/mL, while the MICs for the populations that evolved in the presence of colistin were >512 µg/mL indicating that these populations had acquired colistin resistance. The antibiotic susceptibility to different antibiotics, including colistin, was analyzed in all evolved populations and in the wild-type strain using test-strips. Despite discrepancies between double dilution and test strips, which preclude the use of the latter to inform on clinically relevant antibiotic resistance [34]; test strips can be used in pairwise comparison of wild-type and mutant strains using an operational definition of resistance [35]. Further, it has been described that test strips usually define lower colistin MICs than double dilution in the case of *K. pneumoniae* [36]. Hence, the use of this technique may allow a better discrimination between the levels of colistin resistance as a function of the degree of amplification. Using this methodology, we confirmed that populations that have evolved without colistin showed a similar susceptibility phenotype than the wild-type strain, while the populations that evolved in presence of colistin were less susceptible to colistin than the wild-type strain (Table 1). None of the samples showed changes in susceptibility to other antibiotics besides colistin, suggesting that the genetic changes involved in the phenotype of resistance were specific for colistin.

### 2.2. Colistin Resistance Is Not Associated with Previously Described Resistance Mutations

To analyze the genetic change(s) responsible for the colistin resistance phenotype, four populations that have evolved in the presence of the antibiotic, two control populations that have evolved in absence of antibiotics and the parental strain were re-sequenced, and these re-sequenced genomes were mapped against the already sequenced genome of their parental, wild-type strain. The presence of single nucleotide polymorphisms (SNPs), insertions and/or deletions (indels) is potentially responsible for the colistin resistance phenotype that was analyzed. Some mutations were found in both the control populations and in the populations challenged with colistin. These mutations are involved in the adaption for growing in the experimental conditions and were discarded from further analysis in the evolved mutants since they are not involved in colistin resistance. Data were filtered [37] by quality (probability that a given base is called correctly by the sequencer) and depth (number of reads covering any position), discarding any data with quality < 30 and depth < 20. By using these criteria, 11 possible SNPs were found. Among them, three synonymous substitutions, one conservative mutation, three missense mutations and four mutations localized in intergenic regions were found (Table 2). Seven of these mutations were found only in one evolved population, while four were found in two or more evolved populations. However, no common mutation was found in all evolved populations. Notably, in most cases (including all mutations present in more than one population), the mutations were located in a genomic region that was amplified in all the populations that had evolved in the presence of colistin (see below).

Of the three missense mutations, one was localized in the gene encoding the succinate dehydrogenase cytochrome b 556 subunit, and the prediction of the effect, analyzed with PROVEAN, was neutral (−0.727/Arg76Cys). The other two missense mutations were predicted to have a deleterious effect; one was localized in a gene encoding the phage regulatory protein CII (−4.135/Asp61Asn) and the other in a gene encoding the PTS beta-glucoside transporter subunit IIABC (−5.264/Val502Gly). However, both deleterious missense mutations showed a low number of reads containing them (low depth) and also a low coverage (Table 2), indicating they were not predominant in the studied populations. None of these mutations was present in any of the genes described so far in other works to be involved in the acquisition of colistin resistance.

### 2.3. Chromosomal Amplifications Are Associated with Colistin Resistance

It has been described that unstable tandem gene amplification generates heteroresistance to colistin in *Salmonella typhimurium* [38]. To address if genomic amplifications may be involved in the phenotype of colistin resistance in the evolved populations, we analyzed the sequenced genomes using the CNV-seq tool [39]. Some regions showed an increased number of reads of the genome in the wild-type strain and in all studied populations, controls included, indicating that they are present in more than one copy in the genome of *K. pneumoniae* kp52145. Hence, the increased copy numbers of those regions were not selected under colistin challenge, and they were not further analyzed. In addition to these common regions, a chromosomal amplification of 34,569 bp containing 48 coding sequences (CDS) was identified in the four populations that had evolved in presence of colistin, while this region was not amplified in the populations that had evolved in the absence of antibiotic challenge (Figure 1). To further confirm that this region was amplified in the evolved populations, its copy number was estimated by real-time PCR comparing the amount of two genes, one inside (BN49-RS11300) and another outside (*rpoB*) the putatively amplified region. Statistically significant (*p* < 0.05) differences ranging from 11 copies to 1.5 copies were found in the studied populations (Figure 2), confirming that the amplification of this region is linked to colistin resistance in the studied evolved populations. To further support that the region is amplified, the copy numbers of the genes BN49_RS28910, BN49_RS11385 and BN49_RS11425 were also estimated. In all cases, amplification was detected in all populations, and the observed trend in the level of amplification was the same: 1 > 4 > 3 > 2. Notably, those populations presenting higher copy numbers of the amplified region also display higher colistin MICs (Table 1). This result, together with the absence of any other relevant genetic change, supports that the genetic cause of colistin resistance is the amplification of this genomic region. Four hypothetical proteins, 30 phage-related genes, three domain-containing proteins with unknown function and 11 genes encoding proteins presenting different functions (Figure 3 and Table 3) were identified in this region. Several of the genes present in this region were related to DNA metabolism, three to DNA mobility (integrase, IS1 family transposase and DNA invertase) and five to transcription and replication (RNA polymerase-binding protein DksA, DNA adenine methylase, replication endonuclease, DinI family protein and levan regulatory domain protein).

Two genes encoded proteins analogous to transporters: a membrane protein and a multidrug ABC transporter ATPase. The membrane protein possesses a domain belonging to Phage_holin_2_3 superfamily. This family includes small hydrophobic phage proteins called holins with one transmembrane domain. This gene was followed by another gene homologous to lysozyme, a genomic arrangement (holin-lysozyme) that has been described previously in a phage [41]. The gene encoding the putative lysozyme contains a phage lysozyme domain, whose family includes lambda phage lysozymes and *E. coli* endolysins [42]. Concerning the putative ATPase of an ABC transporter, it contains a protein–protein interaction domain (SEFIR domain-containing protein), which has been associated to transmembrane receptors and soluble factors with eukaryotic origin and with unknown functions [43].

One gene was homologous to *orfA*, which encodes a putative protein present in retron EC67 of *E. coli* [44]. The retron Ec67 is found in a 34 kb sequence flanked by direct repeats of a 26 bp and contains, among other genes, an orf encoding a Dam DNA methylase [44]. Notably, the sequence amplified in the evolved *K. pneumoniae* Kp52145 populations that had evolved in the presence of antibiotics displays similar direct repeats of a 26 bp flanking the amplified sequence (Figure 3), and a gene identified as a DNA adenine methylase was present inside.

It has been described that retron Ec67 is inserted into a prophage between cos sequence and the DNA packaging gene [45]. A similar genetic structure, with a cos sequence in the left side and a terminase-like family protein, was found in the 34 Kb amplified sequence (Figure 3).

The identification of a large number of phage-related genes suggests that this region could be a prophage. The analysis of the complete genome of *K. pneumoniae* kp52145, using the webserver PHASTER, showed the presence of four prophages (Table 4, Appendix A). One of them, the *Enterobacteriaceae* phage Fels-2 [46], corresponds to the sequence that is amplified upon colistin challenge. Blast search of NCBI database (20 December 2020) showed that this phage has been detected in 98 Enterobacterales. Among them, the highest prevalence was for *Klebsiella pneumoniae* with 43 hits followed by *Salmonella enterica* subs. enterica serovar Typhi with 39 hits.

## 3. Discussion

The spread of carbapenemase-producing *Enterobacteriaceae* has increased the use of colistin for treating infections by these organisms. This is the case of *K. pneumoniae*, where resistance to carbapenems is a relevant problem [47], making the use of other antibiotics, such as colistin, tigecycline or aminoglycosides, a clear need. Nevertheless, the use of these antibiotics is also selecting resistance to them. In the case of colistin, the acquisition of *mcr* is a main element in the development of resistance [10]. Nevertheless, genetic modifications, selected in presence of colistin, in the treated patient, may also contribute to such resistance [9,17]. To ascertain those potential modifications, we performed experimental evolution assays for 30 days in the presence of increasing amounts of colistin of the clinical *K. pneumoniae* strain kp52145 (serotype O1:K2) [31]. Notably, genomic analysis of the evolved populations showed that they do not present any of the mutations previously described to be associated with colistin resistance in other pathogens, such as those present in *mgrB*, *phoPQ*, *pmrCAB*, and *crrAB* [9,17]. The only common modification found in the populations that have evolved in the presence of colistin was the amplification of a 34-kb genomic region, which mainly contains phages’ genes. Increased colistin resistance in *Salmonella typhimurium* due to amplification has been described, but in this case, the amplified region included the gene *pmrD*, a positive regulator of the expression of genes that modify lipid A [38]. The facts that amplification is the only genetic modification found, that it is present in all evolved populations and that the populations presenting higher copy numbers are also the ones with higher colistin MICs support that the genetic cause of acquiring colistin resistance is the amplification of this genomic region. However, the biochemical basis of colistin resistance due to the amplification of this region remains elusive. As above stated, the amplified region seems to be a prophage. Some of the genes present in this region are related to transport (membrane protein and multidrug ABC transporter ATPase), a feature that might be linked to the observed resistance phenotype, while another one is annotated as DNA adenine methylase, whose expression may modify antibiotic resistance. Nevertheless, the potential association of the amplification of these genes with colistin resistance remains a speculation.

Antibiotic resistance has been associated with the presence of prophages in some cases. In some cases, this is due to the presence of known resistance genes inside the phage genome [48]. However, there are also examples where the presence of cryptic prophages, not containing known antibiotic resistance genes, has been associated with antibiotic resistance, although the mechanisms involved are largely ignored [49]. A similar situation might be occurring in the present study. Although the molecular bases of resistance are not known yet, our results support that gene amplification can be involved in the acquisition of colistin resistance in *K. pneumoniae* and that retrons and prophages might play a role in this phenotype. Most studies on the genetic basis of antibiotic resistance focus on the acquisition of resistance genes and the selection of mutations. Together with previously published works, our study shows that gene amplification in general and of that of prophages in particular might also play a role in antibiotic resistance that, with some exceptions, is still underestimated [50].

## 4. Materials and Methods

### 4.1. Strain and Culture Medium

The clinical strain *K. pneumoniae* kp52145 (serotype O1:K2) [31], whose sequence is available at https://www.ncbi.nlm.nih.gov/nuccore/ (Reference Sequence: chromosome NZ_FO834906.1; plasmid II, NZ_FO834905.1; plasmid I, NZ_FO834904.1) [51,52], was used for the studies. Bacteria were grown in Mueller Hinton (MH) or Mueller Hinton II (MH II) liquid or agar plates [53] at 37 °C.

### 4.2. Antibiotic Susceptibility

The colistin MICs of the wild-type *K. pneumoniae* kp52145 strains and the evolved populations were determined by double dilution following CLSI rules [54]. The susceptibility phenotype of the evolved *K. pneumoniae* strains was determined as well using Minimal Inhibitory Concentration (MIC) test strips (Liofilchem) in MH II (for colistin) or MH (for gentamycin, erythromycin, tetracycline, chloramphenicol, ciprofloxacin, imipenem, cefoxitin and ceftazidime) agar plates. The conditions for the assay were those recommended by CLSI for disk-diffusion assays [36].

### 4.3. Colistin Evolution Assay

An experimental evolution assay using inhibitory concentrations of colistin in MH II was performed following protocols common in this type of assays [25,26,27,28,29,30]. Previous work had shown that 2 µg/mL polymyxin killed half of the *K. pneumoniae* kp52145 population in one hour [55]. Futhermore, we determined that the same concentration of colistin inhibited *K. pneumoniae* kp52145 growth in our experimental conditions. Hence, this concentration was chosen as the starting point in our assays. The evolution was made in quadruplicate. Four different isolated colonies were independently suspended in 1 mL NaCl 0.85% each. Each suspension was used to inoculate 2 mL MH II, with or without 2 µg/mL colistin (MIC), with approximately 10^3^ CFU/mL each. A total of eight cultures (four with and four without colistin) were then performed. Cultures were grown at 37 °C and 180 r.p.m. Every 24 h, 2 µL of overnight cultures were used to inoculate new 2 mL MH II at a 1:1000 dilution. This step was repeated for 30 days, increasing the concentration of colistin two-fold every five days, with the experiment ending when colistin concentrations reached 64X MIC. Four control cultures were grown in parallel, under the same conditions, but in the absence of colistin. At the end of the evolution assay, the evolved populations were further subcultured (three sequential passages) without antibiotic before analyzing the antibiotic resistance phenotype to avoid the possibility of transient induction of resistance.

### 4.4. Genomic DNA Extraction and Re-Sequencing of the Evolved Populations

Genomic DNA of each evolved sample was obtained from 4 mL of an overnight culture using the G NOME^®^ DNA isolation kit (MP Biomedicals) The quality and quantity of the DNA was determined by agarose gel electrophoresis and by using a NanoDrop spectrophotometer, respectively. The genomes of the wild type strain, two controls (populations 5,6), and four evolved samples (populations 1–4) were re-sequenced at the *Unidad de Genómica Antonia Martín Gallardo, Parque Científico de Madrid*, Spain. To that goal, TruSeq DNA PCR-free Prep (Illumina) based libraries from 5 µg of DNA were obtained and sequenced using MySeq Illumina technology in a single-end 1X150 format. The numbers of reads were: wild type, 1107220; population 1, 1341078; population 2, 1008802; population 3, 1276618; population 4, 1104682; population 5, 1332846; population 6, 1247129. The coverage of each sample was in the 24–45X range.

### 4.5. Bioinformatic Analysis

Sequences were aligned against the genome of the wild-type strain *K. pneumoniae* strain kp52145 deposited at NCBI (https://www.ncbi.nlm.nih.gov/nuccore/; GenBank: FO834906.1). To identify the presence of Single Nucleotide Polymorphisms (SNPs), the programs Bowtie2 [56], SAMtools [57,58], SnpEff [59] and bedtools [60] were used. Quality and depth filters were made with SnpSift [37]. For the search of over-represented sequences, the CNV-seq program [39] was used. Results were visualized using the Integrated Genome Viewer (IGV, Broad institute, http://software.broadinstitute.org/software/igv/). The effect of missense mutations was predicted with the program PROVEAN [61]. Scores ≤ −2.5 are predicted to have a deleterious effect, while scores > −2.5 are predicted to have a neutral effect.

The genes localized in the over-represented region were identified using the information available for the reference sequence of *K. pneumoniae* kp52145 (see above) and by homology with other proteins using Blastx (https://blast.ncbi.nlm.nih.gov/Blast.cgi). The alignment of sequences was performed with Clustal OMEGA [40]

The PHAge Search Tool Enhanced Release (https://phaster.ca/) was used to search for prophages in *K. pneumonia* genome [62,63].

### 4.6. Real-Time PCR

Real-time PCR reactions were prepared using 0.25 ng from the genomic DNA obtained as described above. Primers for *rpoB* reference gene or BN49-RS11300, BN49-RS28910, BN49-RS11385 and BN49-RS11425 genes present in the *K. pneumoniae kp52145* amplified genomic region were used at 10 nM (Table 5). Finally, 10 μL of Power SYBR Green PCR Master Mix (Applied Biosystems, Woolston, Warrington, UK) was added to the mixture. The reaction efficiency for each primer set was estimated on a series of DNA dilutions. Reactions were run on a 7500 Real-Time PCR System (Applied Biosystems, Woolston, Warrington, UK) with the following protocol: 2 min at 50 °C; 10 s at 95 °C; 40 cycles of 10 s at 95 °C; 1 min at 60 °C, followed by a melting curve generated from 60 °C to 95 °C. Ten replicates were performed, and the fold change of BN49-RS11300 copy number was estimated using the 2^−ΔΔCt^ method [64]. The number of copies was estimated as the average of the different determinations. The statistical significance of the results was estimated by *t*-test.

## Figures and Tables

**Figure 1 ijms-22-00649-f001:**
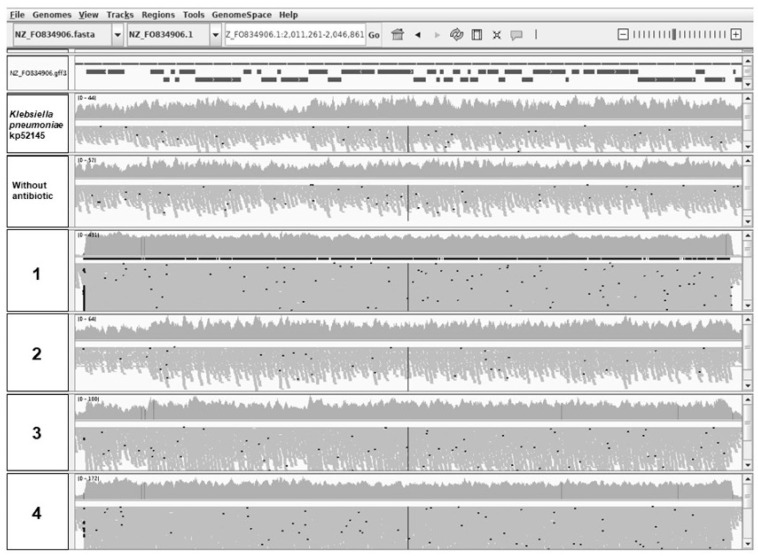
Image of the reads of the putatively amplified region obtained using Integrated Genome Viewer (IGV). An increased number of reads, mainly in populations 1, 3 and 4, which correspond with MICs of 128, 64 and 128 µg/mL, was observed.

**Figure 2 ijms-22-00649-f002:**
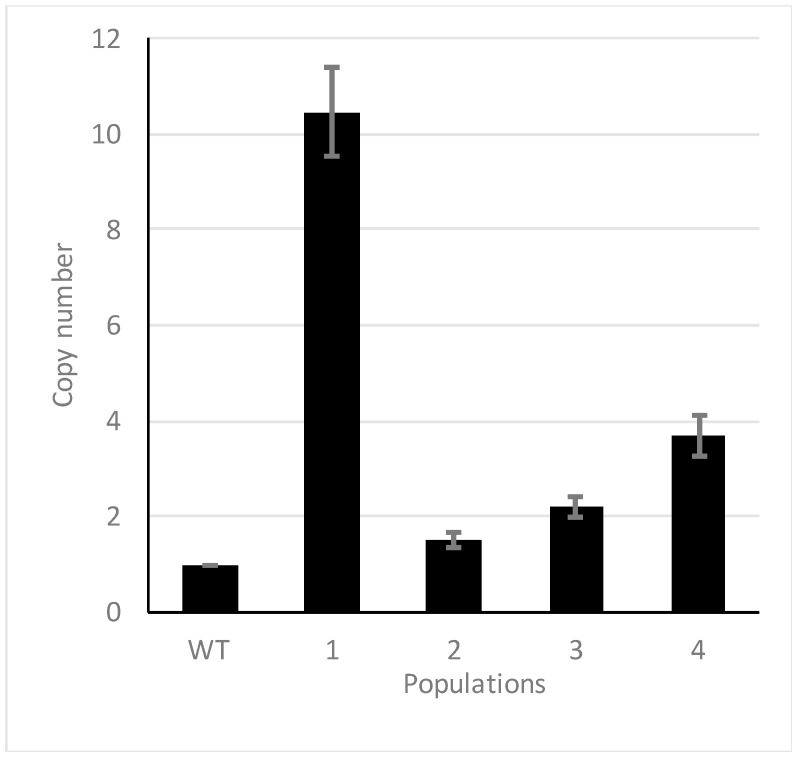
Analysis of the copy number of the amplified region by real-time PCR. Copy number of the amplified region was estimated measuring the amount of BN49-RS11300 present in each population and considering that the wild-type strain presents one copy of the gene. The figure shows the average and standard deviation of copy numbers for each of the populations. In all cases, the differences were statistically significant (*p* < 0.05).

**Figure 3 ijms-22-00649-f003:**
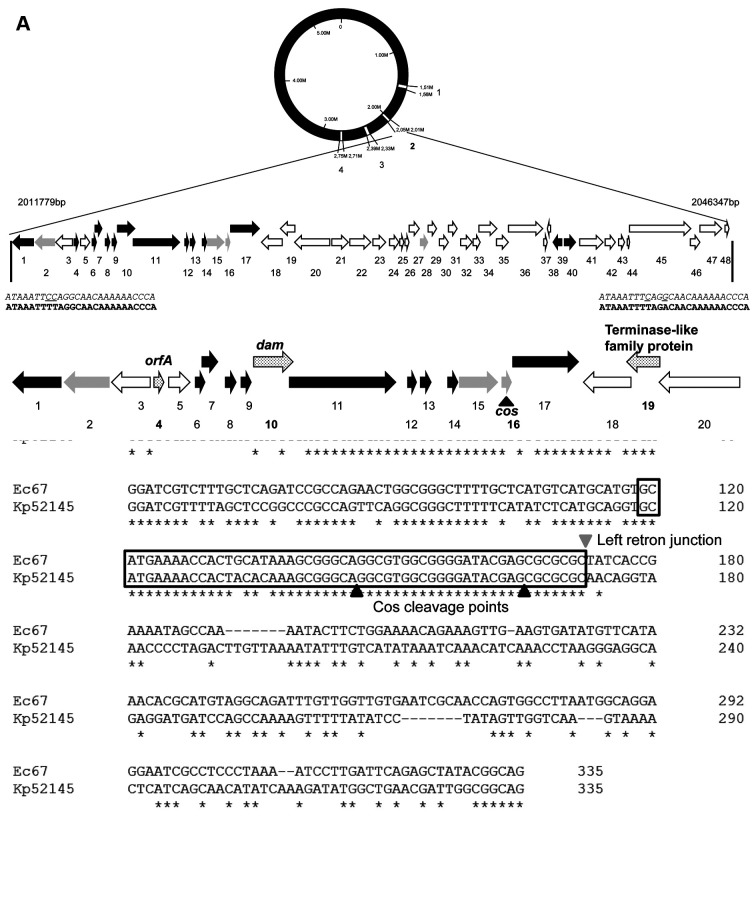
Region in the *K. pneumoniae* chromosome, and genes identified within, that is over-represented in populations that have evolved in presence of colistin. Panel **B** shows the localization of the other four prophages identified in *K. pneumoniae* kp52145. Panel **A** shows the genetic structure of the 34569 bp amplified region, correspondent to prophage 2. Arrows show the orientation of predicted open reading frames ORF. Hypothetical proteins (grey arrows), phage-related proteins (white arrows) and other functions (black arrows). Numbers under arrows are correlated with numbers in Table 3. The figure also shows the 26 bp direct repeats flanking this genomic region sequence. In bold, the sequences present in *E. coli*; in italics, *K. pneumoniae* sequences. The nucleotides that differ between both microorganisms are underlined. Panel B shows the genomic region containing genes identified as related with retron Ec67; *orfA*, *dam* and Terminase-like family protein (dotted arrows), and the position of *cos* sequence (black triangle). **C**, alignment of conservative *cos* site (boxed) and *cos* cleavage points (black triangles) and retron junction (grey triangle) of Ec67 and *K. pneumoniae* kp52145 (Clustal OMEGA [40].

**Table 1 ijms-22-00649-t001:** Minimal Inhibitory Concentrations (MICs) of different antibiotics for *K. pneumoniae* evolved in the presence or in absence of colistin.

Evolution ^a^	Population	MIC (µg/mL) ^b^									Copy Number ^d^
		GM	EM	TC	CL	CI	IP	FX	TZ	CS ^c^	
A	1	0.38	32	2	3	0.064	0.38	4	0.19	**128**	10.46
	2	0.38	48	2	4	0.032	0.75	6	0.19	**48**	1.50
	3	0.5	48	2	4	0.064	0.75	4	0.19	**64**	2.20
	4	0.5	32	1.5	4	0.064	0.5	4	0.125	**128**	3.69
B	5	0.5	48	2	6	0.064	0.5	4	0.125	0.75	1
	6	0.75	48	1	3	0.047	0.25	4	0.125	0.5	1
	7	0.5	32	1.5	4	0.047	0.38	4	0.125	0.5	ND
	8	0.75	48	2	6	0.064	0.5	4	0.125	0.5	ND
C	wt	0.38	32	1.5	4	0.064	0.75	4	0.125	0.5	1

^a^ Evolution condition in MH II. A, with 64 XMIC colistin; B, without colistin and C, wild-type *K. pneumoniae* kp52145 without evolution. **^b^** GM, gentamycin; EM, erythromycin; TC, tetracycline; CL, chloramphenicol; CI, ciprofloxacin; IP, imipenem; FX, cefoxitin and TZ, ceftazidime and CS, colistin. ^c^ In bold, MICs higher than the wild-type *K. pneumoniae* kp52145 without evolution. ^d^ corresponds to the fold-change in the copy number of the amplified region analyzed by real-time PCR. ND: not determined.

**Table 2 ijms-22-00649-t002:** Mutations found in populations of *K. pneumoniae* kp52145 that have evolved in the presence of colistin.

Position a	Mutation		Type b	Population c	Quality	Coverage (%) d	DP e	Product/Blast Homology
**1852616**	C > T	Arg76Cys	MSM	3	225	68.97	29	Succinate dehydrogenase cytochrome b556 subunit
**2011786**	C > T	intergenic_region	IR	1	225	66.66	84	3′ tyrosine-type recombinase/integrase
**2014814**	C > A	Leu48Leu	SS	1	222	90.42	355	Phage repressor protein CI
				3	225	56.25	51	
				4	213	27.27	99	
**2014977**	T > C	intergenic region	IR	1	222	89.23	362	5′ Phage repressor protein CI
				4	225	74.58	118	
**2015036**	A > G	intergenic region	IR	3	225	59.52	42	5′Regulator
**2015480**	G > A	Asp61Asn	MSM	3	225	51.90	80	Phage regulatory protein CII
**2037216**	T > C	Val505Ala	CM	3	225	37.87	66	Phage tail protein
				4	225	67.57	111	
**2043437**	C > T	Ala715Ala	SS	3	225	53.57	56	Phage tail tape measure protein
				4	225	73.74	99	
**2045994**	C > T	intergenic region	IR	1	222	89.71	418	5′ transcriptional activator Ogr/delta
3994455	A > C	Val502Gly	MSM	2	152	40.9	22	PTS beta-glucoside transporter subunit IIABC
5429220	C > T	Glu601Glu	SS	1	222	97.5	40	DEAD/DEAH box helicase

^a^ In bold, mutations localized inside the region that is amplified in the evolved populations. ^b^ Type of mutation that causes the single nucleotide polymorphism (SNP). SS, synonymous substitutions; MSM, missense mutation; CM, conservative mutation and IR, intergenic regions. ^c^ Number corresponds with evolved population in supra-inhibitory concentration of colistin (Table 2). ^d^ Percentage of reads with mutation in relation to total reads. ^e^ Number of reads.

**Table 3 ijms-22-00649-t003:** Genes identified in the over-represented sequence.

Number ^a^	Locus	Position		Strand	Size (bp)	Product/Blast Homology
		First	End			
1	BN49_RS11210	2011880	2012932	−	1053	Tyrosine-type recombinase/integrase
2	BN49_RS11215	2013019	2014008	−	990	Hypothetical protein
3	BN49_RS28910	2014019	2014957	−	939	Phage repressor protein CI
4	BN49_RS28915	2015046	2015267	+	222	Putative uncharacterized protein ORFA in retron EC67 (*E. coli*)/ Regulator (*E. coli*, *K. pneumoniae*)
5	BN49_RS11225	2015300	2015809	+	510	Phage regulatory protein CII
6	BN49_RS11230	2015817	2016017	+	201	DUF2724 domain-containing protein
7	BN49_RS11235	2015981	2016322	+	342	DUF5347 domain-containing protein
8	BN49_RS11240	2016390	2016623	+	234	DUF2732 domain-containing protein
9	BN49_RS11245	2016623	2016850	+	228	TraR/DksA family transcriptional regulator
10	BN49_RS11250	2016847	2017704	+	858	DNA adenine methylase
11	BN49_RS11255	2017701	2020115	+	2415	Replication endonuclease
12	BN49_RS11260	2020269	2020457	+	189	levan regulatory domain protein
13	BN49_RS11265	2020468	2020701	+	234	DinI family protein
14	BN49_RS11270	2020988	2021206	+	219	Multidrug ABC transporter ATPase (*E. coli*)
15	BN49_RS11275	2021206	2022048	+	843	Hypothetical protein
16	BN49_RS11280	2022058	2022267	+	210	Hypothetical protein
17	BN49_RS11285	2022264	2023697	+	1434	SEFIR domain-containing protein
18	BN49_RS11290	2023732	2024772	−	1041	Phage portal protein
19	BN49_RS11295	2024769	2025494	−	726	Terminase-like family protein
20	BN49_RS11300	2025494	2027260	−	1767	Terminase ATPase subunit
21	BN49_RS11305	2027403	2028236	+	834	Phage capsid protein
22	BN49_RS11310	2028253	2029311	+	1059	Phage major capsid protein, P2 family
23	BN49_RS11315	2029315	2029965	+	651	Terminase endonuclease subunit
24	BN49_RS11320	2030061	2030525	+	465	Head completion/stabilization protein
25	BN49_RS11325	2030525	2030728	+	204	Tail protein X
26	BN49_RS11330	2030732	2030947	+	216	Membrane protein (*Enterobacteriaceae*)
27	BN49_RS11335	2030967	2031440	+	513	Lysozyme
28	BN49_RS11340	2031442	2031819	+	378	Hypothetical protein
29	BN49_RS11345	2031816	2032244	+	429	LysB family phage lysis regulatory protein
30	BN49_RS11355	2032319	2032771	+	453	Tail completion protein R
31	BN49_RS11360	2032764	2033210	+	447	Phage virion morphogenesis protein
32	BN49_RS11365	2033279	2033857	+	579	Phage baseplate assembly protein V
33	BN49_RS11370	2033854	2034213	+	360	Baseplate assembly protein
34	BN49_RS11375	2034200	2035108	+	909	Baseplate assembly protein
35	BN49_RS11380	2035101	2035706	+	606	Phage tail protein I
36	BN49_RS11385	2035703	2037424	+	1722	Tail fiber protein/phage tail protein
37	BN49_RS11390	2037424	2037606	+	183	Phage tail protein
38	BN49_RS28920	2037587	2037739	−	153	Tail fiber assembly protein
39	BN49_RS11395	2037760	2038207	−	448	IS1 family transposase (Pseudo: partial start) DDE_Tnp_IS1 superfamily
40	BN49_RS11400	2038403	2038969	+	567	DNA invertase
41	BN49_RS11405	2039112	2040284	+	1173	Major tail sheath protein
42	BN49_RS11410	2040294	2040809	+	516	Phage major tail tube protein
43	BN49_RS11415	2040864	2041166	+	303	Phage tail assembly protein
44	BN49_RS11420	2041181	2041300	+	120	GpE family phage tail protein
45	BN49_RS11425	2041293	2044370	+	3078	Phage tail tape measure protein
46	BN49_RS11430	2044367	2044852	+	486	Bacteriophage tail protein
47	BN49_RS11435	2044849	2045949	+	1101	Phage late control D family protein
48	BN49_RS28925	2046040	2046258	+	219	Transcriptional activator Ogr/delta

^a^ Number correspond with numbers in Figure 3.

**Table 4 ijms-22-00649-t004:** Localization of the prophages identified in the genome of *K. pneumoniae* kp52145. Number 2 corresponds with the over-represented region in evolved mutants.

	Region Length	Genome Position	Condition ^a^
1	50 kb	1506010–1556027	intact
2	34.5 kb	2011787–2046330	intact
3	33.1 kb	2346519–2379687	intact
4	50.6 kb	2700266–2750911	intact

^a^ Prediction if the prophage is intact or incomplete.

**Table 5 ijms-22-00649-t005:** Sequences of the primers used for qPCR.

Name	Sequence (5′-3′)
RpoB-fwd	GATCCGTGGCGTGACTTATT
RpoB-rev	GCCCATGTAGACTTCTTGTTCT
RS11300-fwd	CAGGCCATGCTGCTGTACTT
RS11300-rev	GTCCAGCGGCCCATAGT
RS28910-fwd	CCTTCTGCGGTAACTCCAATAG
RS28910-rev	CAGCTCTGTAGCCCAACTTAAA
RS11385-fwd	CTAGCTCAGTTGGTTCGTTTATTTC
RS11385-rev	GTCGGGCTATCGCTGTTATT
RS11425-fwd	GGAGGTGCGTGATCGTATTG
RS11425-rev	GGCATCTTCCATGCTGGTATAG

## Data Availability

Data sharing not applicable.

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
