# Peer review of "The Acquisition of Colistin Resistance Is Associated to the Amplification of a Large Chromosomal Region in Klebsiella pneumoniae kp52145"

_ijms, 2021, doi:10.3390/ijms22020649_

Round 1

Reviewer 1 Report

This manuscript describes the role of amplification of 34 Kb (3 prophages and 1 prophage questionable) in the development of colistin resistance in K. pneumoniae.

Four colistin mutants were obtained from the WT strain kp51245. These strains were sequenced, RT-PCR and bioinformatic analysis also were performed.

This study adds new knowledge to the field on the colistin resistance in Klebsiella pneumoniae, suggesting a new possible mechanism for the development of colistin resistance.

I have following comments for the author’s consideration,

Major comments:

  1. CLSI states that ‘the only approved MIC method for testing (colistin) is broth microdilution. Disk diffusion and gradient diffusion methods should not be performed.’ EUCAST also recommends the same with this antibiotic. MICs to colistin should be redetermined according to these recommendations.
  2. The authors performed RT-PCR as a confirmation of the amplification of the 34 kb region. However, they only tested one gene from this region, while it presents 48 CDS. It would be necessary to know the expression of at least three more genes (BN49_RS28910, BN49_RS11385 and BN49_RS11425) to understand their role in more detail; because these genes present mutations in several of the most ColR strains (essentially B3 and B4).

Minor comments:

Keywords: Change ‘Stenotrophomonas maltophilia’ by ‘Klebsiella pneumoniae

Introduction:

  1. Page 3 line 5, Change ‘Gram negative’ by ‘Gram-negative’
  2. Page 3 line 10, Change ‘baumanii’ by ‘baumannii
  3. Page 3 line 10, It is more appropriate to use the new taxonomic term ‘Enterobacterales’ instead of ‘Enterobacteriaceae’.
  4. Page 3 line 22, It should be specified that there are different mcr genes reported to date and that mcr-1 is the most widespread.
  5. Page 4 line 1, ‘Gram-negative’
  6. Page 4 lines 12-14, mutations in lpx genes are more common in baumannii while mutations in phoPQ, pmrAB and others are more common in Enterobacterales. It is necessary to mention it.
  7. Page 4 lines 19-25. The last part of the introduction should explain the novelty of the study, the objective of the study, the hypotheses or be used as a brief approach to what the reader will find in the work. I recommend rewriting these lines, since they are more similar to the conclusions of the work than to the last part of the introduction.

Material and methods

  1. Page 5 line 3. The authors have used the strain kp51245, instead of other more studied strains such as ATCC 13883 or ATCC 700721. What was the reason for choosing this strain?
  2. Page 5 line 16. I recommend mentioning previous works in which this technique has been described, if it is the case. What was the reason to start with a concentration of 2 µg/ml when MIC is 0.5 µg/ml?
  3. Page 5 line 19. CFU/mL would be more appropriate.
  4. Page 5 line 23. Add that during the experiment the authors worked with a total of 8 tubes (4 with colistin and 4 without colistin, controls). The experiment is already well described, but to clarify this I think can be helpful to the readers.
  5. Page 6 line 11. Although previously mentioned, the name of the strain and the reference sequence codes must be indicated.
  6. Page 6 line 17. Delete ‘equal’.
  7. Page 6 line 23. Insert link to the website.
  8. Page 7 line 1-6. Separate and rewrite this sentence, it is very long.
  9. Page 7 line 3. Why was the gene BN49-RS11300 selected from all 48 CDS?
  10. Page 7 line 13. ‘t-test’.

Results

  1. Page 8 lines 8-11. This sentence is unclear. Have the authors seen mutations common to all strains? Which strain was this compared to? This sentence must be rewritten.
  2. Page 8 line 11. As with ‘depth’ specify what is ‘quality’.
  3. Page 10 line 1. Delete first ‘,’.
  4. Page 10 line 19 and 21. Put this reference like the other references.

Discussion

  1. It is necessary to add more information of the 4 prophages identified. If there is only information about the phage Fels-2 indicate what is known about it. Indicate if it is very common and in what percentage of the total strains of Enterobacterales sequenced in genbank appears.

Tables:

  1. Table 1. Specify that the primers were used for real-time PCR.
  2. Table 1. Change ‘RS-rv2’ by ‘RS-rv’.
  3. Table 2. Specify a single MIC value for the WT strain to CI and IP antibiotics.
  4. Table 2. I recommend to the authors that the populations 1,2,3 and 4 of group A be designated as 5,6,7 and 8 for example. I think it can be misleading with table 3.
  5. Table 3. Add in a new column the type of mutation that causes the SNP (the 4 classes indicated in lines 13-15 on page 8).
  6. Table 4. Indicate which regions would correspond to each of the 4 prophages.

References: Scientific names in italics.

Author Response

This manuscript describes the role of amplification of 34 Kb (3 prophages and 1 prophage questionable) in the development of colistin resistance in K. pneumoniae.

Four colistin mutants were obtained from the WT strain kp51245. These strains were sequenced, RT-PCR and bioinformatic analysis also were performed.

This study adds new knowledge to the field on the colistin resistance in Klebsiella pneumoniae,suggesting a new possible mechanism for the development of colistin resistance.

Answer: we appreciate the positive opinion of the referee concerning our work

I have following comments for the author’s consideration,

Major comments:

  1. CLSI states that ‘the only approved MIC method for testing (colistin) is broth microdilution. Disk diffusion and gradient diffusion methods should not be performed.’ EUCAST also recommends the same with this antibiotic. MICs to colistin should be redetermined according to these recommendations.

Answer: It is true that for colistin the MICs for clinical definition of resistance must be performed by microdilution. We have made these determinations and in all cases MIC increases from 2 mg/L in the wild-type to >512 in the mutants. This information is now included (Page 8, lines 22-24). However, besides clinical definition of resistance, ECOFF-based definitions (based as well in breackpoints) and operational definition (not based in breakpoints, just in pair-wise comparison of the wild-type and the mutant) have been used (see reference 52). Since colistin MICs determined by E-test are usually lower than double-dilution MICs (see reference 53), and E-test allows a better discrimination of low differences we have kept the E-test information, in coherence with the data of sensitivity to other antibiotics. This is better explained in Page 9, lines 1-8.

  1. The authors performed RT-PCR as a confirmation of the amplification of the 34 kb region. However, they only tested one gene from this region, while it presents 48 CDS. It would be necessary to know the expression of at least three more genes (BN49_RS28910, BN49_RS11385 and BN49_RS11425) to understand their role in more detail; because these genes present mutations in several of the most ColR strains (essentially B3 and B4).
  2.  

Answer: Following referee's recommendation, we have checked if these genes are amplified. They are and follow the same trend than the gene already tested (Page 11, lines 13-16). We have also double-checked the presence of mutations in these genes and others known to be involved in colistin resistance and did not find any mutation. One aspect that we were no aware is that several ColR strains present mutations in these prophage genes and have been able to find the papers describing these mutations. Could please the referee provide the corresponding references? This would be an added value for the discussion of our article.

Minor comments:

Keywords: Change ‘Stenotrophomonas maltophilia’ by ‘Klebsiella pneumoniae

Introduction:

  1. Page 3 line 5, Change ‘Gram negative’ by ‘Gram-negative’

Answer: Done

  1. Page 3 line 10, Change ‘baumanii’ by ‘baumannii

Answer: Done

  1. Page 3 line 10, It is more appropriate to use the new taxonomic term ‘Enterobacterales’ instead of ‘Enterobacteriaceae’.

Answer: Done

  1. Page 3 line 22, It should be specified that there are different mcr genes reported to date and that mcr-1 is the most widespread.

Answer: Done

  1. Page 4 line 1, ‘Gram-negative’

Answer: Done

  1. Page 4 lines 12-14, mutations in lpx genes are more common in baumannii while mutations in phoPQpmrAB and others are more common in Enterobacterales. It is necessary to mention it.

Answer: Done

  1. Page 4 lines 19-25. The last part of the introduction should explain the novelty of the study, the objective of the study, the hypotheses or be used as a brief approach to what the reader will find in the work. I recommend rewriting these lines, since they are more similar to the conclusions of the work than to the last part of the introduction.
  2. Answer: Done

Material and methods

  1. Page 5 line 3. The authors have used the strain kp51245, instead of other more studied strains such as ATCC 13883 or ATCC 700721. What was the reason for choosing this strain?

We agree with reviewer that are other strains could be used. However, kp51245, firstly described in 1986, has also been used in several studies. Further. it belongs to the most clinically relevant K. pneumoniase phylogenetic group and its genome sequence (not just the draft genome but the closed genome) is available, a condition required for our work studies. Besides, we knew for previous publications that 2 mg/L of colistin killed half of the population of this strain in one hour, and this concentration was a good starting point for our evolution experiments. This information is now included in page 6, lines 3-4.

  1. Page 5 line 16. I recommend mentioning previous works in which this technique has been described, if it is the case. What was the reason to start with a concentration of 2 µg/ml when MIC is 0.5 µg/ml?

Some more references have been included here and in the introduction. Concerning the concentration used, we understand this is confusing. However, as rightly stated by the referee in the general comments, colistin MIC values are different when measured bytest stripsor by double dilution, and for evolution we used the MIC determined by double dilution, which inhibits bacterial growth in the experimental conditions of the evolution. This is better explained in Page 6, lines 2-6 and in Page 9, lines 1-8.

  1. Page 5 line 19. CFU/mL would be more appropriate.

Answer: Done

  1. Page 5 line 23. Add that during the experiment the authors worked with a total of 8 tubes (4 with colistin and 4 without colistin, controls). The experiment is already well described, but to clarify this I think can be helpful to the readers.

Answer: Done

  1. Page 6 line 11. Although previously mentioned, the name of the strain and the reference sequence codes must be indicated.  

Answer: Done

  1. Page 6 line 17. Delete ‘equal’.

Answer: Done

  1. Page 6 line 23. Insert link to the website. .

Answer: Done

  1. Page 7 line 1-6. Separate and rewrite this sentence, it is very long

Answer: Done

  1. Page 7 line 3. Why was the gene BN49-RS11300 selected from all 48 CDS?

Being in the middle of the amplified region, we thought this gene would be the better candidate to confirm the variation in copy number observed from the analysis of whole-genome sequencing. Nevertheless, the suggested analysis of three more genes present in different regions of the prophage confirms that the region is amplified (Page 11, lines 13-16).

  1. Page 7 line 13. ‘t-test’.

Answer: Done

Results

  1. Page 8 lines 8-11. This sentence is unclear. Have the authors seen mutations common to all strains? Which strain was this compared to? This sentence must be rewritten.

Yes, some mutations were found in the strains evolved without antibiotic and in the strain evolved. All strains were compared with the sequence available in database (NCBI) including the wild type used in this work. The sentence has been rewritten. See page 9, lines 21-25.

  1. Page 8 line 11. As with ‘depth’ specify what is ‘quality’.

Answer: Done

  1. Page 10 line 1. Delete first ‘,’.

Answer: Done

  1. Page 10 line 19 and 21. Put this reference like the other references.

Answer: Done

Discussion

  1. It is necessary to add more information of the 4 prophages identified. If there is only information about the phage Fels-2 indicate what is known about it. Indicate if it is very common and in what percentage of the total strains of Enterobacterales sequenced in genbank appears.

Answer: More information on the structure and the localization of the other phages (Table 5, supplemental Table S1) and the prevalence of Fels-2 is included (Page 13, lines 5-8). Figure 3 has also modified to include the position of these phages within the bacterial genome

Tables:

  1. Table 1. Specify that the primers were used for real-time PCR.

Answer: Done

  1. Table 1. Change ‘RS-rv2’ by ‘RS-rv’.

Answer: Done

  1. Table 2. Specify a single MIC value for the WT strain to CI and IP antibiotics.

Answer: Done

  1. Table 2. I recommend to the authors that the populations 1,2,3 and 4 of group A be designated as 5,6,7 and 8 for example. I think it can be misleading with table 3.

Answer: We kept these numbers, but following referee's suggestions we numbered consecutively; 5,6,7 and 8 the populations of group B

  1. Table 3. Add in a new column the type of mutation that causes the SNP (the 4 classes indicated in lines 13-15 on page 8).

Answer: Done

  1. Table 4. Indicate which regions would correspond to each of the 4 prophages.

Answer: Done. A new table with the information has been included (Table 5)

  1. References: Scientific names in italics.

Answer: Done

Reviewer 2 Report

The work is devoted to the study of resistance of one of the most important human pathogens K. pneumoniae to colistin which is the antibiotic of “last resort”. The Authors encountered an interesting of the genome feature of the resistant strain which they created and decided to be the first to describe the observation.

This is certainly worth understanding. However, it does not negate the fact that the work is preliminary in nature. This is indirectly evidenced by the Authors themselves when they often use such phrases as “seems to be”, “unlikely” “might play” etc. but directly follows from the fact that the Authors do not know how revealed gene amplification is associated with phenotypic resistance (Page 12, lines 12-13).

In this situation, the minimum that needs to be done is to evaluate the expression of these genes of bacteria that grow in a medium without colistin and with the introduction of colistin at subinhibitory concentrations.

I would like to note the brevity in the description of the methods used in the work. This is not appropriate. For example:

  1. Antibiotic susceptibility: describe in more detail the method of setting this test (according to what protocol, cultivation conditions, volume, and density of the suspension were used, etc.)
  2. Genomic DNA extraction: there are no data on methods for determining the quantity and quality of isolated DNA; What was the protocol for making DNA libraries? what instrument was used for sequencing?
  3. Bioinformatic analysis: What programs were used to assemble and annotate genomes?
  4. Data on the results of bioinformatic processing of genomes should be included in additional files (Raw reads;  Genome size, bp; Number of contigs;  GC content; Number of genes; Protein-coding sequences; tRNAs)

Minor remarks

  1. Apparently, Stenotrophomonas maltophilia was added to keywords by mistake.
  2. Inconsistency in the spelling of Gram-negative / gram negative through the text. Check it and reach the uniformity.
  3. Uncover SNPs abbreviation in that place when it comes first (Material and methods)
  4. There are grammatical errors, I recommend proofreading the text.
  5. Significant number of literature sources are over 5 years old. There is no uniformity in the Reference List and Citations Style.

Author Response

The work is devoted to the study of resistance of one of the most important human pathogens K. pneumoniae to colistin which is the antibiotic of “last resort”. The Authors encountered an interesting of the genome feature of the resistant strain which they created and decided to be the first to describe the observation.

Answer: We appreciate the positive opinion of the referee concerning our work

This is certainly worth understanding. However, it does not negate the fact that the work is preliminary in nature. This is indirectly evidenced by the Authors themselves when they often use such phrases as “seems to be”, “unlikely” “might play” etc. but directly follows from the fact that the Authors do not know how revealed gene amplification is associated with phenotypic resistance (Page 12, lines 12-13).

Answer: We appreciate the point of the referee. Since no other change besides amplification is observed and the degree of amplification correlates with the resistance level, we are confident that we know the genetic basis of resistance: the amplification. However, it is true that the biochemical basis of resistance remains elusive. This is not so rare in the studies on mutation-driven resistance. In occasions, as target or transporter mutations the biochemical aspects are obvious, but in other cases, as the recently described mutations in fusA, quite frequent in clinical and in vitro obtained P. aeruginosa antibiotic resistant - to different antibiotics- are not so clear. However, the description of these mutations opens the field in two directions: one is to provide information on antibiotic resistance mutations, not yet known, that could be introduced in molecular epidemiology studies and the other to study novel mechanisms of resistance. The fact that the genetic basis or resistance are known, but the information on the biochemistry behind is still elusive is better discussed in Page 11, lines 18-19 and in Page 14, lines 5-7.

In this situation, the minimum that needs to be done is to evaluate the expression of these genes of bacteria that grow in a medium without colistin and with the introduction of colistin at subinhibitory concentrations.

Answer: We appreciate the point, but the referee would like to take into consideration that the expression of most antibiotic resistance genes is not induced by the antibiotic they inactivate, hence it is doubtful that this approach will provide insights in the biochemical bases of resistance (as above stated and better explained now in the article, the genetic basis of resistance are clear: the amplification of a specific genomic region). Further, sub-MIC concentrations produce changes in the expression of several genes, not related with antibiotic resistance, to the point that Julian Davies and ourselves proposed independently that antibiotics may have signaling functions, more than inhibitory ones, in nature (See Int J Med Microbiol,2006 Apr;296(2-3):163-70; Proc Natl Acad Sci U S A. 2006 Dec 19;103(51):19484-9). In the case of colistin, although induction of resistance by this antibiotic was reported in 1977 (ntimicrob Agents Chemother 1977 Jul;12(1):1-3.), there is not any further relevant information on such potential induction published later on. Further, recent work shows that sub-MIC colistin selects mutants, it does not induce transient resistance (see J Med Microbiol. 2020 Apr;69(4):521-529). In the case of mutation/genetic modifications, which before the discovery of mcr-1were thought to be the only cause of resistance to this antibiotic, it is the mutation itself what provides resistance; induction is not needed and does not happen. As the consequence of some of the mutations, the expression of differen genes may change, but this is in the absence of antibiotoc  (see for instance references 11, 13, 14, 18 in this respect). Even when transcriptomic studies are performed (see reference 18), these studies are made in the absence of colistin, because it is the mutation, not the presence of the antibiotic what renders resistance. In fact, to avoid any potential problem of transient induction in our experimental setup, we re-cultured the populations in the absence of colistin (three passages) and resistance remained, indicating that induction by colistin is not needed to achieve the phenotype.

I would like to note the brevity in the description of the methods used in the work. This is not appropriate. For example:

  1. Antibiotic susceptibility: describe in more detail the method of setting this test (according to what protocol, cultivation conditions, volume, and density of the suspension were used, etc.)

Answer: MICs were performed in agreement with CLSI rules. This information is now included as well as the corresponding references.

  1. Genomic DNA extraction: there are no data on methods for determining the quantity and quality of isolated DNA; What was the protocol for making DNA libraries? what instrument was used for sequencing?

Answer: Thanks for the query, since the information was really short. More detailed information is now included

  1. Bioinformatic analysis: What programs were used to assemble and annotate genomes?
  2. Data on the results of bioinformatic processing of genomes should be included in additional files (Raw reads; Genome size, bp; Number of contigs;  GC content; Number of genes; Protein-coding sequences; tRNAs)

Answer: We did not sequence novel isolates. We re-sequenced mutants derived from a strain that has been already sequenced. Since we were not clear enough, the feature that we re-sequenced the mutants is now stated. Hence, assembling and annotation is not needed, because the original genome is already assembled and annotated. We have mapped the reads over the known genome and looked-for changes using common tools for this type of analysis that are indicated, with the corresponding references, in the methods section. Since the mutants have the same genome (excepting the few mutations and the amplifications found), the numbers of genes, Protein-coding sequences and tRNAs are the same and genome size and GC content is very similar; the only small changes are due to the amplified region. Information on the number of reads is now included (Page 7, lines 2-4).  

Round 2

Reviewer 1 Report

The manuscript has improved considerably and the authors have taken into account all my recommendations. I only have some minor comments due to text errors (regarding the file "Download manuscript"):

  1. Check that all scientific names are in italics. Kp52145 without italics.
  2. Line 302-303 page 13, delete “Minimal Inhibitory Concentration”. MIC must be defined the first time it appears in the manuscript.
  3. Line 305 page 13, delete “by”. Line 202 page 8, change “Panel B” by “Panel A”.
  4. I would recommend that the position and size of the tables and figures be more in accordance with the format. This will enhance the visual quality of the manuscript.
  5. Regarding the recommendations of my previous review of this manuscript, the authors performed RT-PCR of the 3 requested genes (BN49_RS28910, BN49_RS11385 and BN49_RS11425). I just wanted to clarify that the reason I recommended RT-PCR of these 3 genes was because SNPs were observed in populations 3 and 4, also population 1 showed an SNP for the first gene. Throughout the manuscript the authors successfully demonstrate that these mutations are not involved in colistin resistance and that all the genes from 34 Kb sequence are amplified equally in each population.

Reviewer 2 Report

I am satisfied with the responses of the authors and the corrections they made.